# Cortical gradient of a human functional similarity network captured by the geometry of cytoarchitectonic organization

Yao Meng [1,2], Siqi Yang[1,2], Jinming Xiao[1,2], Yaxin Lu[1,2], Jiao Li[1,2], Huafu Chen[1,2] & Wei Liao [1,2 ✉]

Mapping the functional topology from a multifaceted perspective and relating it to underlying cross-scale structural principles is crucial for understanding the structural-functional relationships of the cerebral cortex. Previous works have described a sensory-association gradient axis in terms of coupling relationships between structure and function, but largely based on single specific feature, and the mesoscopic underpinnings are rarely determined. Here we show a gradient pattern encoded in a functional similarity network based on data from Human Connectome Project and further link it to cytoarchitectonic organizing principles. The spatial distribution of the primary gradient follows an inferior-anterior to superior-posterior axis. The primary gradient demonstrates converging relationships with layer-specific microscopic gene expression and mesoscopic cortical layer thickness, and is captured by the geometric representation of a myelo- and cyto-architecture based laminar differentiation theorem, involving a dual origin theory. Together, these findings provide a gradient, which describes the functional topology, and more importantly, linking the macroscale functional landscape with mesoscale laminar differentiation principles.

[1] The Clinical Hospital of Chengdu Brain Science Institute, School of Life Science and Technology, University of Electronic Science and Technology of China, Chengdu 611731, P. R. China. [2] MOE Key Lab for Neuroinformation, High-Field Magnetic Resonance Brain Imaging Key Laboratory of Sichuan Province, University of Electronic Science and Technology of China, Chengdu 611731, P. R. China. ✉email: weiliao.wl@gmail.com

The human brain is characterized by heterogeneous patterns of structural wiring and functional connectivity (FC). Typically, the FC is defined as the correlation of two elements' blood oxygen level-dependent (BOLD) time series[1]. The end result, the functional brain network, represents the organization of neural activity and thus provides profound insights regarding macro-scale functional specialization and integration[2,3]. However, the FC itself cannot provide organizing principles of cortical topology.

Increasing evidence has implied that a set of anatomically distributed functional systems/networks were anchored on the cerebral cortex by some axes describing the spatially graded changes in the expression of connectivity patterns, which were the so-called "Gradients"[4–6]. These spatial architectures inferred from BOLD fluctuations have established substantial relationships with microscopic gene expressions[7] and neurotransmitter profiles[8,9], mesoscopic cytoarchitecture and cortical morphology[10–13], and macroscopic functional system hierarchies[4] and dynamics[14–16], which were also necessarily intertwined with cognitive function, behaviors, and brain-related disorders[17–20] that illustrated the importance of studying cross-scale interactions among the genetic, molecular, cellular, and macroscale levels of brain circuitry and connectivity and behavior. These exemplary cross-scale interaction studies provide a unified framework, which consists of a continuous varying axis to describe the spatial organization of the cerebral cortex from multiple perspectives. Previous studies usually focused on single-feature based similarity matrix embeddings (gradients), which provided one specific aspect of the topology of cerebral functional activities. However, there still lacked a multifaceted informative macroscale description of the cerebral functional topology.

Comprehensively describing the functional characteristics of the cerebral cortex requires local activities and global communication indicators. Brain regions possess various functional properties in many aspects, such as the local activity and global communication relationships, which cannot be elucidated using a single index. Converging multiple descriptive features to identify cross-region relationships, which are involved in cortical morphological research[21], may provide a complementary understanding of the functional topology that is presently and predominately based on single feature association. Metrics derived from spontaneous BOLD fluctuations can describe the functional topology of the cerebral cortex from local and global perspectives. Local fluctuation properties such as the amplitude of low frequency fluctuations (ALFF) and fractional ALFF (fALFF)[22,23] describe the frequency spectrum power, which suggests the energy of neuro-vascular fluctuations, and the regional homogeneity (ReHo)[24–26] quantifies the degree of connections of a given node with its nearest neighbors. When complemented with local fluctuation properties, network model metrics based on graph theoretical indexes abstract the global cortical communication framework into a simplified graph[27,28], such as, integration (degree centrality, DC; global efficiency, gEfficiency; and shortest path length), and segregation (local efficiency, lEfficiency). These network measures capture multifaceted properties of topology of functional interactions among nodes or brain regions under a global framework. Combining the mentioned metrics from global and local scales may therefore provide a more comprehensive representation of the cortical function topological landscape.

An extensively explored topic is the relationship between functional activity with the cortical structural basis[29]. One of the widespread and recognized theories is that functions arise from the structural and differential coupling properties across the cortical mantle[29]. Numerous studies have characterized the associations of BOLD activities and derived the functional network topology with macroscale cortical geometry and morphological features, as well as diffusion signal-based structural networks[30,31]. There have been some spatially more detailed measures, such as myeloarchitecture, cytoarchitecture, and cortical-cortical connections from tract tracings. This converging evidence has been summarized as an evolutional, developmental, converged cortical structural organization principle, the dual origin theory[32], which has not been connected to macroscale functional topology. However, to what extent and manner the structural architecture inferred from mesoscopic cytoarchitectonic information restricts or determines the topological landscape of function still needs to be studied.

Here, we converged multifaceted functional regional activity and network metrics to represent the functional topology of the cerebral cortex and characterized the similarities of topological features. We then leveraged a canonical dimensionality reduction algorithm to map the primary gradient (Fig. 1). We showed the spatial pattern of the primary gradient across the neocortex and tested the reproducibility in internal and external validations. Similar spatial associations with the functional similarity gradient were revealed across the cytoarchitecturally-defined cortical layer, including microarray-detected gene expression levels and layer thickness distributions. The geometry using the framework of the dual origin theory modeled the primary gradient. Collectively, this work revealed a low dimensional embedding (gradient) of functional topology, and identified the associations with micro- and mesoscale structural features. More importantly, these findings demonstrated a dual origin principle of the primary gradient alignment with the theoretical hypothesis derived from cytoarchitectonic studies.

## Results

**Cortical gradient of functional similarity networks**. We mapped the cortical functional similarity networks, integrated with multifaceted functional measures, including regional spontaneous fluctuations and global topological properties from the resting-state BOLD functional MRI (fMRI) signal in a large-sample ($N = 999$) multimodal dataset, which was the Human Connectome Project (HCP). All the results presented in the main text is based on the data of first session of HCP (HCP-REST1), unless otherwise stated. We defined network nodes ($N = 360$) with a multi-modal parcellation scheme[33], and the edges on the basis of the similarities between the functional metrics of two nodes. These metric maps (including ALFF, fALFF, ReHo, DC, shortest path length, lEfficiency, and gEfficiency) spatially vectorized and z-scored for each participant (Fig. 1a). These metrics reflected different aspects of the functional topology of the cerebral cortex, mainly from the local scale fluctuation characteristics and global communication descriptive features of the network model. Each parcel's topological metrics formed a feature vector, which described area topological characteristics from a more comprehensive point of view. To quantify the similarities of functional topologies of nodes, we computed normalized angles across the feature vectors of nodes to define the node-to-node (parcel-to-parcel) similarities, to provide functional similarity matrices[6,21,34] (Fig. 1b). The encoded functional similarity matrices were averaged across participants, to yield a group-level similarity matrix. To acquire the principal spatial variation pattern of this group-level topological similarity matrix, we deployed a well-recognized dimensionality reduction algorithm, called diffusion map embedding, to map the embedded spatial patterns or "gradients"[35]. The first component of the embeddings, the primary gradient, explained over half of the variance (the primary gradient, $54 \pm 2\%$; Fig. 1c), which represented a dominant spatial pattern of the functional similarity network. The subsequent

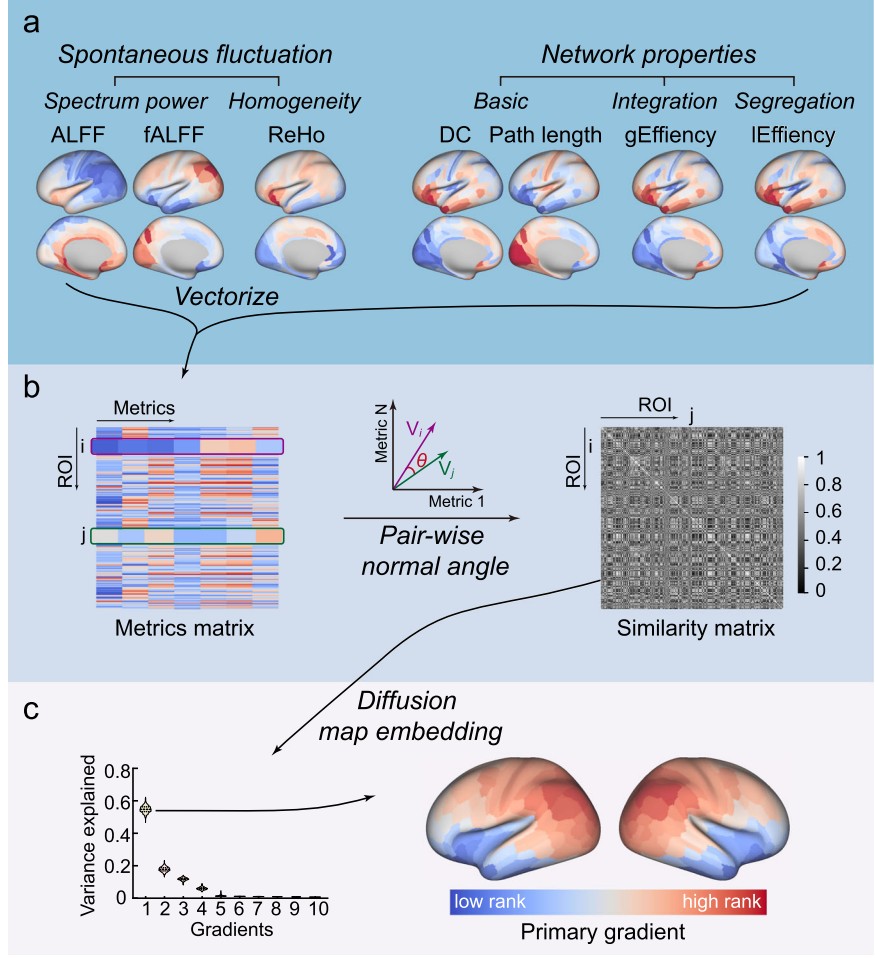

**Fig. 1 Schematic diagram of the multifaceted functional gradient. a** shows seven metrics derived from the resting-state BOLD-fMRI signal. For each individual, all seven metrics maps vectorized to form a feature matrix, and the normal angle of each pair of brain areas or region of interest was calculated as a similarity measure (**b**). A symmetric similarity matrix resulted from preceding procedures fed into a diffusion map embedding algorithm, to project the high dimensional similarity profile to a series of low dimensional embeddings. The first component of embeddings was selected as the primary gradient because it better explained the half variance of the input similarity matrix and was then rendered in the bottom row on the inflated surface (**c**).

analyses were then mainly based on this primary gradient because of its dominant place across embeddings.

The primary gradient indicated a nonuniform spatial distribution, and showed gradual changes along the allocortex-isocortex axis. The primary gradient generally displayed a gradual increase from the inferior-anterior to the superior-posterior of the cerebral cortex, with the lowest located in the temporal pole and paralimbic regions, and the highest in the parietal-occipital junction area (Fig. 2a). It could be reproduced in internal validation (HCP-REST2 dataset) ($r_s = 1$, $p_{SAC} < 0.0001$) and external validation [independent Midnight Scan Club (MSC) dataset] ($r_s = 0.53$, $p_{SAC} = 0.0006$) analysis (Supplementary Fig. 1). The primary gradient was be reproduced across different parcellation schemes (Supplementary Fig. 2). The proposed gradient differed from the canonical functional connectome gradient, which represented a unimodal-transmodal axis (Supplementary Fig. 3, $r_s = -0.35$, $p_{SAC} = 0.04$). We found a correlation ($r_s = 0.90$) between the temporal signal-to-noise ratio (tSNR) map and the primary gradient, Supplementary Fig. 4. Moreover, we found that the tSNR map spatial correlated with all functional metrics maps except for the ReHo map (Supplementary Table 1), consistent with previous studies[36,37]. We also compared the primary gradient and tSNR map (both z-scored) using paired *t*-test. The difference is widely spread across the cortex, including the

occipital cortex, lateral temporal cortex, prefrontal cortex, orbitofrontal cortex, and insula (Supplementary Fig. 4b). Although the primary gradient and the tSNR maps show a significant correlation, the discrepancy suggests the primary gradient of the cortical functional similarity network is not simply dominated by tSNR.

We dissected the brain into four types according to a histologically defined atlas based on cortical laminar differentiation classes[38]. The paralimbic class remained at the lower end relative to the other three cortical classes, which had more diverse distributions (Kruskal–Wallis test, $\chi^2_{(3)} = 132.9$, $p < 0.0001$) (Fig. 2b, paralimbic: $-0.24 \pm 0.07$; heteromodal: $0.03 \pm 0.16$; idiotypic: $0.06 \pm 0.12$; unimodal: $0.07 \pm 0.14$) across the primary gradient (axis), suggesting a diverging role of paralimbic regions in the dominant embedding of topology similarities. The converged result was shown in the distributions of primary gradient in terms of functional network identities, which also showed a limbic system apparently dissociated from other networks (Supplementary Fig. 3b).

**Layer-specific characteristic of functional similarity embedding.** We used transcriptomic and histology data to determine the layer-specific characteristic of the primary gradient.

## a. Multifaced primary gradient

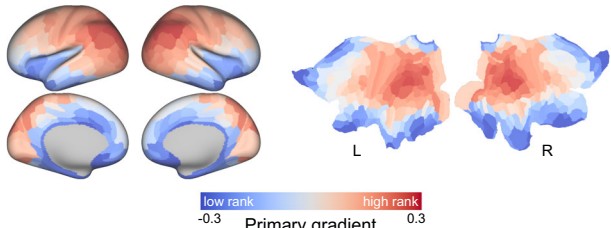

## b. Distribution across laminar differration classes

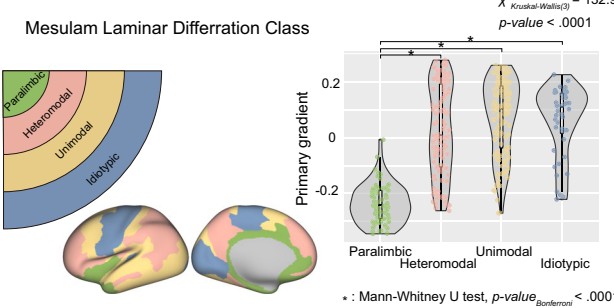

**Fig. 2 Space distribution of the multifaceted functional gradient. a** The first component of embeddings, namely the primary gradient, was rendered on the inflated surface (left) and unfolded flat surface (right). **b** According to the Mesulam laminar differentiation class atlas, the primary gradient was classified as a different class based on the corresponding spatial location.

First, we used layer-specific gene expression maps from the Allen Human Brain Atlas (AHBA)[39]. This publicly available *post-mortem* human brain transcriptional atlas contains brain-wide gene expression data measured with microarrays. We acquired the layer-specific gene expression map from a previous study, which grouped related genes into sets representative of supragranular (Layers 1–3), granular (Layer 4), and infragranular (Layers 5, 6) layers[40,41]. To characterize the relationships between the macroscopic functional topology gradient with microscopic specific layer-related gene expressions, we correlated these two maps using Spearman's rank correlation coefficient, and defined the statistical significance level using a spatially constrained null model[42]. The primary gradient showed distinct associations with granular and infragranular layers (Fig. 3a), which had a strong positive correlation with the granular layer 4 gene expression ($r_s = 0.65$, $p_{SAC} = 0.003$), but not with the infragranular layers 5 and 6 ($r_s = -0.71$, $p_{SAC} = 0.001$). The supragranular layer related genes expression map did not have a significant association with the primary gradient ($r_s = -0.05$, $p_{SAC} = 0.76$).

To assure distinctive separation relationships with granular and infragranular layers in the preceding gene expression correlation analysis, we used a histological brain atlas, the BigBrain[43], to characterize the relationships with the cytoarchitectonic-defined layer-specific cortical thicknesses. We quantified the thickness of the supragranular, granular, and infragranular layers and parcellated these maps using the same scheme in the original BigBrain space. We then correlated these layer-specific thickness maps with primary gradients across the cerebral cortex. The results showed a similar pattern compared to the preceding gene expression analysis, with the granular layer thickness positively correlated (Fig. 3b, $r_s = 0.47$, $p_{SAC} = 0.01$) with the primary gradient with an opposite trend ($r_s = -0.18$, $p_{SAC} = 0.27$), which was shown in the infragranular layer thickness. In addition, the supragranular layer thickness did not correlate with the primary gradient ($r_s = -0.09$, $p_{SAC} = 0.47$).

We then determined the relationship of the primary gradient relation with an in vivo microstructure profile gradient (MPC), which was embedded in the covariance pattern of the surface-depth dependent T1/T2 myelination profile across the cortical mantle[44]. The MPC showed differential patterns of myelination along the depth of the cortical surface, which provided insight into microstructural patterning across different layers. The high rank in the MPC tended to show a more uniform myelination profile across different layers. By contrast, the high rank of the MPC represented a nonuniform pattern, with the granular layer with more myelination. The primary gradient negatively correlated with the MPC (Fig. 3c, $r_s = -0.61$, $p_{SAC} = 0.002$), which showed the functional organization axis associated with a gradual transition from uniform myelination to mid-surface preference myelination.

**Association with the "dual origin" theory**. The studies of numerous neuroscientists regarding the cytoarchitecture of the human cerebral cortex and corticocortical connections using tract-tracing in non-human primates over the past two centuries have resulted in a comprehensive framework for interpreting the structural organization of the cortex, namely the "dual origin" theory. According to this theoretical principle, the cerebral cortex has evolved from two primordial allocortical moieties: the paleocortex (piriform cortex) and archicortex (hippocampus). We established the distinguishing associations between the primary gradient with cytoarchitecture-defined layer-specific microscopic gene expressions and mesoscopic cortical thicknesses across three layers in a previous section. Here, we evaluated the relationships between the primary gradient and geodesic distance from the paleocortex (piriform cortex) and the archicortex (hippocampus). We found a positive association in both geodesic distance maps with the primary gradient (Fig. 4a, Paleocortex: $r_s = 0.77$, $p_{SAC} = 0.0007$ and Archicortex: $r_s = 0.62$, $p_{SAC} = 0.01$). We used these two geometry maps to construct a linear regression model to fit the primary gradient. The proposed model explained 88% variance (Fig. 4b, $F_{(2, 357)} = 1361$, $p < 0.0001$, adjusted $R^2 = 0.88$) in the primary gradient and the two geodesic distance maps significantly predicted the gradient (Paleocortex: $F_{(1, 357)} = 899.8$, $p < 0.0001$; Archicortex: $F_{(1, 357)} = 1514$, $p < 0.0001$).

## Discussion

Our findings revealed a novel cortical organization axis or gradient, which was embedded in functional similarity networks and captured the geometry organizing principle obtained from cytoarchitecture studies. The proposed gradient showed cortical layer-specific characteristics in gene expressions and layer thicknesses. With-in dataset repeated sessions and independent datasets were then used to confirm the stability and reproducibility of the gradient pattern.

Understanding the functional topology principle of the cerebral cortex is a fundamental question in the field of neuroscience. We proposed a multifaceted approach to assess the functional arrangement on the cortical mantle based on BOLD neurovascular coupling signals in vivo. By aggregating local fluctuation characteristics, which included energy implied spectrum power (ALFF and fALFF) and regional homogeneities, along with global network metrics describing functional segregation (local efficiency) and integration (global efficiency, shortest path length, and degree centrality), we showed the proposed functional similarity matrices/networks encoded multiple topological features and were a general representation of the cerebral functional landscape. This type of procedure already exists in brain morphological studies[6,21,34,45–47]. Compared with previous functional connectome/network studies[48–50], which represented the

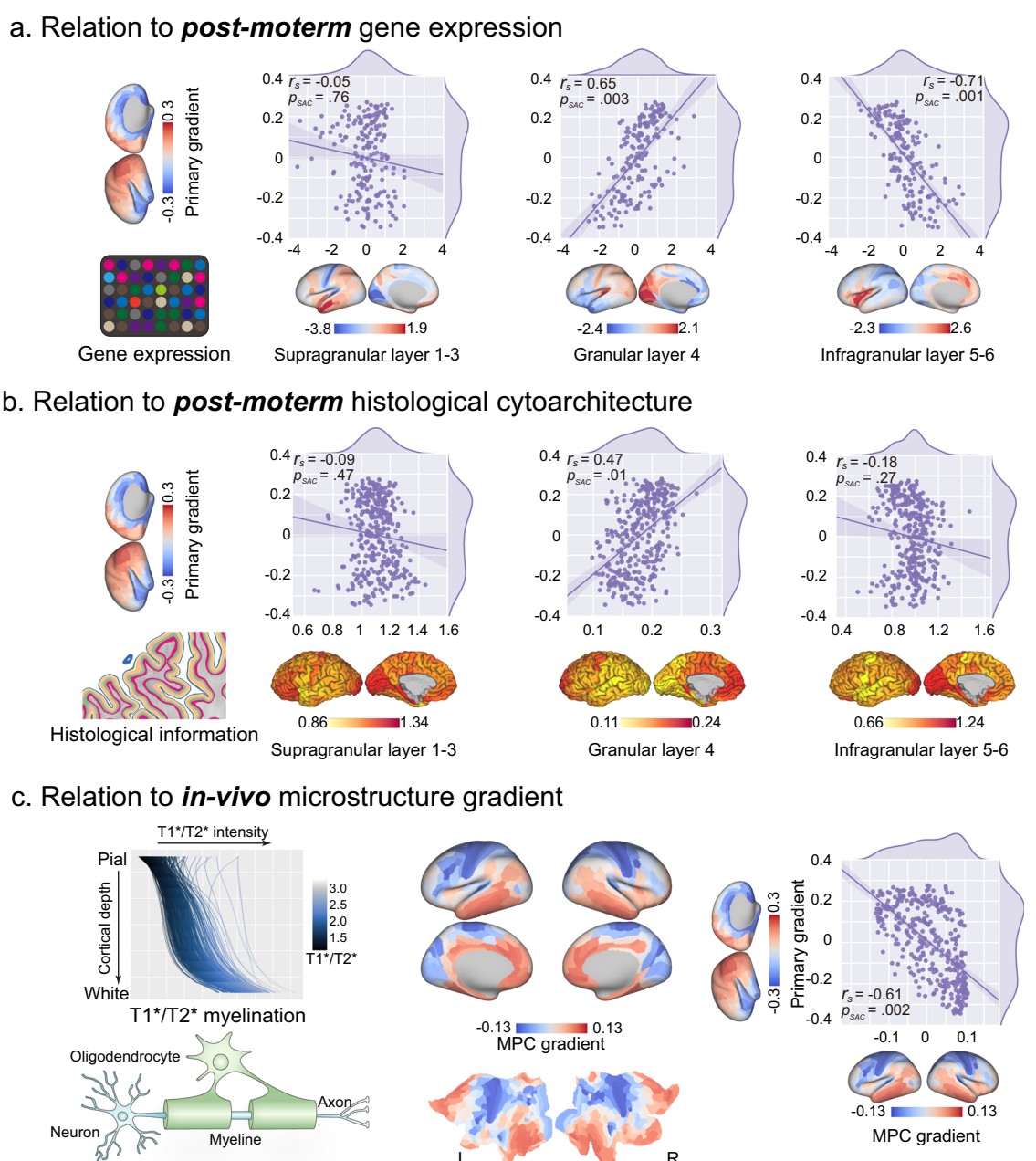

**Fig. 3 Layer-specific characteristics of the primary gradient. a** The primary gradient differentially correlated with layer-related gene expression, involving no correlation with the supragranular layer (layers $1 − 3$) ($r_s = −0.05$, $p_{SAC} = 0.76$), but strong positive correlation with the granular layer (layer 4) ($r_s = 0.65$, $p_{SAC} = 0.003$), and strong negative correlation with the infragranular layer (layers 5, 6) ($r_s = −0.71$, $p_{SAC} = 0.001$). **b** The primary gradient differentially correlated with layer thickness (mm), but had no correlation with the supragranular layer (layer $1 − 3$) ($r_s = −0.09$, $p_{SAC} = 0.47$). A strong positive correlation with the granular layer (layer 4) ($r_s = 0.47$, $p_{SAC} = 0.01$) and negative correlation with the infragranular layer (layers 5, 6) ($r_s = −0.18$, $p_{SAC} = 0.27$) was also found. **c** The primary gradient showed a negative association with the axis described by the microstructure profile covariance gradient, which depicted the surface depth-dependent myelination (T1/T2) profile covariation acquired from Paquola et al.[44] MPC, microstructure profile covariance.

organization of neural activity by computing the similarity of BOLD fluctuation[2], the functional similarity network in the present study was based on the similarities of high-dimension topological characteristics. This approach may provide more general principles of functional topology.

By capturing the low dimensional representation of each area in an abstract features space, we obtained the landscape of functional topology based on multiple metrics. This spatial representation was nonuniformly distributed and appeared to continuously vary across the cortical mantle. The proposed

gradient pattern represented an inferior-anterior to superior-posterior variation axis across the cortex surface, which likely implied the transition of the allocortex to isocortex. Differing from multiple converged evidence of a sensory-association hierarchical axis[51] manifest in human cortical anatomy[40,52], function[53–55], connectivity[4], evolution[56–59], and development[51], we proposed an alternative cortical axis that represented the functional topology inferred from multiple metrics, which may enhance our knowledge of the associations between macroscale function with mesoscopic cytoarchitecture.

## a. Relation to **"dual origin"** theory

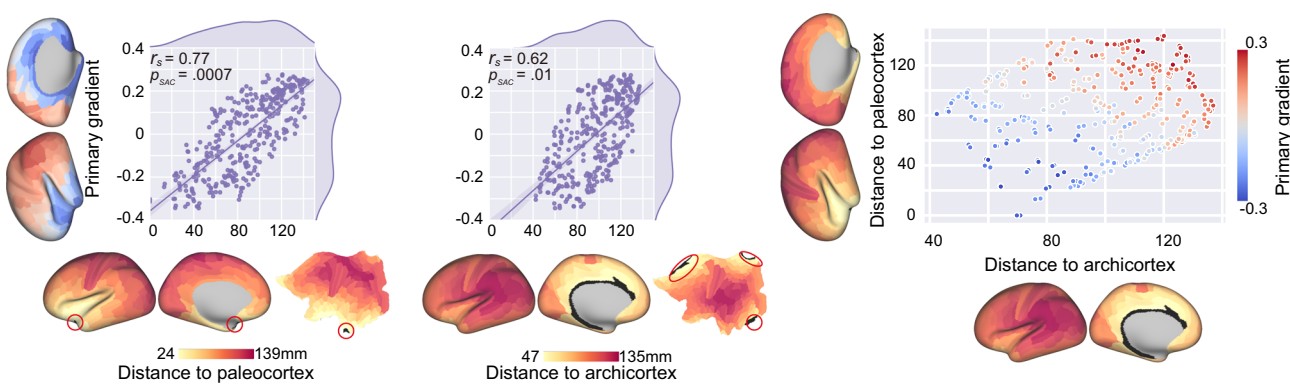

## b. Cytoarchitectonic model predictes functional similarity gradient

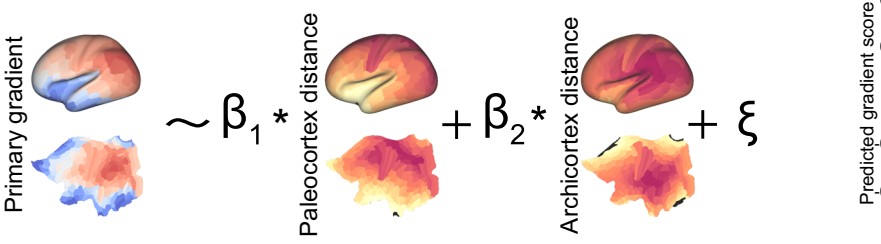

**Fig. 4 The primary gradient and the cytoarchitectonic dual origin theory. a** The primary gradient positively correlated with both paleo- and archicortex geodesic distance maps (Paleocortex: $r_s = 0.77$, $p_{SAC} = 0.0007$; Archicortex: $r_s = 0.62$, $p_{SAC} = 0.01$) and varied from the bottom-left to upper-right in the coordinate system defined by two distance maps. Each distance map was a mean geodesic distance map, which contained the averaged distance of all seed distance maps of the region of interest (colored in black). **b** A linear regression model of two distance maps largely explained the primary gradient (adjusted $R^2 = 0.88$, $p < 0.0001$).

In our findings, the primary gradient showed differential associations to different cortex layers in both microarray gene expressions and histological layer thicknesses. The primary gradient showed strong positive relationships to thicknesses and expressions of related genes of the granular layer, and also the fourth layer in the canonical six-layer laminar structure. However, the supragranular layer (layers 1−3) did not show any associations with the primary gradient, in terms of related gene expressions and thicknesses. In contrast with the granular layer, the expressions of related genes and thicknesses of infragranular layers (layers 5, 6) showed negative correlations or corresponding trends with the primary gradient. These differential findings suggest the layer-specific characteristics of the proposing primary gradient and its possible linkage with underlying cytoarchitecture. The granular layer receives afferent connections from the infragranular layer, which is involved in a feedforward system. In contrast, the infragranular layer is the major origin of reciprocal feedback connections, which preferentially terminate at the first layer[60,61]. Our findings may link the primary gradient with corticocortical connection hierarchy and the spatial distribution of feedforward and feedback connections.

The laminar differentiation of human cortex is least evident in allocortical areas, which have three cortex layers and mainly include the archicortex (hippocampus) and paleocortex (pyriform cortex) with more differentiation developed in the periallocortex, which is adjacent to and surrounding the allocortical regions, on the proisocortex[32]. The laminar differentiation stream eventually reaches the isocortex (neocortex), which

clearly shows six cortex layers. The laminar differentiation stream has two major branches, the dorsal and ventral trends. The dorsal trend originates from the archicortical allocortex, while the ventral trend streams from another allocortex (the paleocortex)[32]. The aforementioned principles of differentiation streams have been included in a theorem, namely the dual origin theory[62,63]. The geometry of this theory on the cortical surface can be simply expressed as the geodesic distance to two seeds, the paleocortex and archicortex. In the present study, the primary gradient showed strong correlations to both paleo- and archicortex distance maps, which presumably represented the geometry principle of the dual origin theory[11]. Surprisingly, the topological primary gradient was well-modeled and predicted by a linear model (combination) of the distance maps. These findings demonstrated the close relationship between the functional gradient and cytoarchitecture structures, and suggested that the functional topology may be largely controlled by the laminar differentiation stream.

How human functional topology from a multifaceted view is distributed across the cortical mantle and related to underlying multiscale structural features is a challenging question in the field of neuroscience. The present results proposed a novel spatial variation pattern-gradient, which represented the functional topology from multiple perspectives and was displayed as an allocortex-isocortex transition axis. Our findings only elucidated the linkage between functional gradients with mesoscale structural principles from a statistical aspect, so the detailed mechanism supporting this across-scale and modality relationship remains to be studied.

## Methods

### MRI data

*Human Connectome Project.* We utilized resting-state BOLD-fMRI datasets from the HCP S1200 release 3T MRI for conducting the analyses. Original data release included 1,113 healthy young adults. We excluded subjects who failed to complete the scan sessions (less than four resting-state fMRI scan sessions, $N = 95$) and a batch of incorrectly preprocessed subjects ($N = 19$) by the HCP. Finally, we enrolled 999 young, healthy adults (female = 541, age = $26.95 \pm 3.47$ years) from the HCP S1200 release for whom all four rs-fMRI and structural scans were available. All MRI data used in this study were publicly available from HCP's Connectome Database (ConnectomeDB, https://www.humanconnectome.org/software/connectomedb). Participant recruitment procedures and informed consent forms, including consent to share deidentified data, were previously approved by the Washington University Institutional Review Board as part of the HCP. Briefly, we utilized preprocessed rs-fMRI data acquired from the HCP. Original data went through the minimally preprocessing pipeline[64], aligned to the fs_LR32k group space using Multimodal Surface Matching All area feature-based registration (MSM-All)[65]. Data denoising was achieved by FMRIB's independent component analysis-based X-noiseifier (ICA-FIX)[66]. We used data of both two runs in HCP-REST1 session to conduct the main analysis.

In addition, we used the data of both two runs in HCP-REST2 session as internal validation to test the reproducibility of the primary gradient.

*Midnight Scan Club.* We used an independent dataset-Midnight Scan Club (MSC) as the external validation of the proposed functional similarity network encoded multifaced gradient. The detail descriptions for the scan parameters, subject inclusions, and imaging preprocessing pipeline can be found in Gordon et al.[67]. Briefly, ten healthy adults were scanned at Washington University using a 3T Siemens Trio scanner (Siemens, Campbell, CA, USA). The study was approved by the Washington University School of Medicine Human Studies Committee and Institutional Review Board, and informed consent was obtained from all participants. Participants completed 12 scanning session on 10 sequential days. Ten rs-fMRI sessions were collected using gradient-echo EPI sequence (run duration = 30 min, TR = 2,200 ms, TE = 27 ms, flip angle = 90°, 4-mm isotropic voxel resolution) with eyes open. All sessions underwent slice timing correction and were normalized to a whole brain mode intensity value of 1000. Images then underwent distortion correction, motion correction (frame-wise displacement > 0.2 mm censored), demeaning and detrending, multiple regression (including whole brain, ventricular and white matter signals, and motion regressors derived by Volterra expansion), and band-pass filtering (0.009 Hz < $f$ < 0.08 Hz). Then, a BOLD-fMRI volumetric time series (both resting-state and task) were sampled to each subject's original mid-thickness left and right-hemisphere surfaces using the ribbon-constrained sampling procedure, and deformed and resampled from the individual's original surface to the 32k fs_LR surface.

### Temporal and topological feature computation of rs-fMRI data.

For constructing the functional similarity matrices/networks, we computed a set of metrics, which included the local fluctuation metrics and global network metrics. We used a well-recognized multimodal parcellation atlas (MMP)[33] to resolve the preprocessed BOLD-fMRI time series and local fluctuation metrics from vertex to parcel levels.

*Spontaneous fluctuation.* The ALFF was computed as the averaged square root of each frequency across $0.01 - 0.08$ Hz in the BOLD-fMRI time series' power spectrum[22]. fALFF was defined as the ratio of the power of each frequency at the low frequency range (0.01–0.08 Hz) to that of the entire frequency range[23].

*Regional homogeneity.* ReHo was defined as Kendall's coefficient concordance of a given vertex's time series with its closet neighbors[24].

*Functional network and network metrics.* There were three fundamental perspectives to describe the network model, including integration (degree centrality, shortest path length, and global efficiency), and segregation (local efficiency)[68]. DC reflected the numbers of neighbors connected to the node, which determined the importance of the given node in the network. lEfficiency was the fraction of node's neighbors that were also neighbors of each other, which quantified the ability for specialized processing to occur within densely interconnected groups of brain regions, namely functional segregation. The ability to rapidly combine specialized information from distributed brain regions functional integration was quantified by the shortest path length (path length) to its neighbor and the deriving gEfficiency. Each parcel's time series was defined as the spatial mean of all included vertexes' time series. We then correlated each parcel's time series using Pearson's correlation coefficient to obtain the functional connectivity matrix. We used the GRETNA toolbox[69] to compute multi-graph theoretical metrics, the detail computation process and formulas refer to Wang et al[69]. A series of thresholds (0.1–0.3, 0.02 stepwise) was used to control the sparsity of the connectivity matrix. The area under the curve of the network metric-sparsity characteristic curve was computed and used in subsequent analyses.

### Construction of the cortical functional similarity matrix.

Each metric map was z-scored to normalize the data. For one parcel of the cortex, seven metrics comprised the feature vector. We used normalized angles to define the similarities across parcels, which computed the cosine distance between parcels' feature vectors and transformed to angle representation. Across-parcel similarity matrix was constructed for each participant.

### Cortical gradient computation.

We averaged the all-individual's similarity matrices to yield a group-level similarity matrix. This group-level matrix was submitted to a non-linear dimensionality reduction algorithm, which was called diffusion map embedding. The algorithm was controlled with two parameters, α and $t$, where α controlled the influence of density of sampling points on the underlying manifold (α = 0, maximal influence; α = 1, no influence), and $t$ controlled the scale of eigenvalues of the diffusion operator. We set α at 0.5 and $t$ at 0, a setting that maintained the global relationships between data points in the embedded space, and was more robust to noise in the similarity matrix.

### AHBA transcriptional data.

Layer-specific gene expression profiles were acquired from *Burt* et al.[40] and are openly available to the public via the BALSA database. The human gene expression data were obtained from the AHBA (http://human.brain-map.org). The detailed processing information is described in Burt et al.[40].

### Cytoarchitecture data.

Layer-specific cortical thickness was obtained from the BigBrain database[43] (https://bigbrainproject.org). The areal level thickness map was obtained by parcelling the original layer-specific thickness using a transformed MMP atlas in the *BigBrain* space.

### Microstructure profile covariance gradient.

The MPC gradient was acquired from Paquola et al.[44]. The detail processing information can be found elsewhere[44]. Briefly, based on 110 healthy unrelated young adults (female = 66, age = $28.8 \pm 3.8$ years) of the HCP S1200 release, 14 equi-volumetric surfaces between the outer and inner cortical surfaces were constructed and T1/T2 values systematically sampled to linked vertices from the outer to the inner surface across the whole cortex. A covariance matrix was constructed based on the myelination profiles across cortical surfaces, and a diffusion map embedding algorithm was deployed to extract the dominant component-MPC gradient.

### Statistics and reproducibility.

The correlation between topological gradient and other cortical features was quantified using the Spearman's rank correlation coefficient. Significance was determined by comparing empirical correlation values with the spatial-autocorrelation accounted null model, which was comprised of surrogate maps generated by a spatial-lag model[42]. The linear regression model was constructed using two geodesic maps, which seeded at the paleocortex (piriform cortex) and the archicortex (hippocampus) to fit the primary gradient.

### Reporting summary.

Further information on research design is available in the Nature Research Reporting Summary linked to this article.

## Data availability

All data needed to evaluate the conclusions in the paper are present in main text and the Supplementary Materials. The source data underlying Figs. 2, 3 and 4 are provided as Supplementary Data 1. MRI data used in this study were publicly available from HCP's Connectome Database (ConnectomeDB, https://www.humanconnectome.org/software/connectomedb). The human gene expression data were obtained from the Allen Human Brain Atlas AHBA ("Complete normalized microarray datasets", https://human.brainmap.org/static/download)). Layer-specific cortical thickness was obtained from the BigBrain database (https://bigbrainproject.org).

## Code availability

All the code is openly available at https://github.com/YaoMeng94/FSN-Gradient.

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

## Acknowledgements

This work is supported by National Key Project of Research and Development of Ministry of Science and Technology (2018AAA0100705), National Natural Science Foundation of China (61871077, 62036003, 82202250, and U1808204), Chinese National Science & Technology Pillar Program (2022YFC2009906), Excellent Youth Foundation of Sichuan Scientific Committee (2020JDJQ0016), and China Postdoctoral Science Foundation (BX2021057, and 2022M710615). We thank International Science Editing (http://www.internationalscienceediting.com) for editing this manuscript.

## Author contributions

Y.M., and W.L. designed research; Y.M., S.Y., J.X., Y.L., and J.L. performed research; Y.M., S.Y., and J.X. contributed new reagents/analytic tools; Y.M., and J.X. analyzed data; Y.M. wrote the paper; S.Y., H.C., and W.L. edited the paper.

## Competing interests

The authors declare no competing interests.
