## [Peer Review File · Communications Biology]

Reviewers' comments:

Reviewer #2 (Remarks to the Author):

Thank you for the opportunity to review "Cortical gradient of a functional similarity network captured by the geometry of cytoarchitectonic organization" by Meng and colleagues. The manuscript describes an extension of previous work looking at how functional and cytoarchitectonic gradients are correlated. The main novelty of the work is that the authors include multiple derivatives of fMRI time series, include ALFF, ReHo, etc. and apply the standard diffusion map embedding algorithm to recover gradients. They then show that these are often correlated with other structural gradients, and broadly resemble the Dual Origin pattern.

Overall, the paper is rigorous and comprehensive. The work builds on recent reports and will be of interest to that community. I do, however, have some suggestions for improvement:

1. The narrative is clear, but the writing is nevertheless awkward and needs work. For example:

"The primary gradient demonstrated layer-specific characteristics in microscopic gene expression and mesoscopic layer thickness, and was captured by geometric representation of a myelo- and cyto-architecture based laminar differentiation theorem, involving a dual origin theory."

"decode ubiquitous cognitions"

"Atypical FC defines connectivity on the basis of the similarity between two elements' dynamics"

"continuous transition relationships"

2. In addition, the writing is often more complex than it needs to be. For example

"We then leveraged a canonical embedding algorithm based on manifold learning".

Surely the authors could have just said "dimensionality reduction" rather than "manifold learning".

3. Importantly, the gradient derived by the authors (shown in Fig. 1) is also reminiscent of the well-known tSNR map for fMRI. Given that the gradient is derived entirely from fMRI measures, the authors should verify whether the two are correlated.

4. "All the code is openly available at <https://github.com/YaoMeng94/FSN415> Gradient." Contrary to this statement, there is no such repository and the code is not openly available.

5. How were FC weights transformed to lengths for computing path lengths in the global efficiency measure?

6. The fMRI measures chosen are somewhat arbitrary. What is the rationale for not including other commonly used measures, such as measures of variability (SD, MSSD, MSE, etc.)?

Reviewer #3 (Remarks to the Author):

This work connects macroscopic functional connectivity organizational principles derived from a variety of FC measures (ReHo, ALFF, fALLF) as well as graph theoretical measures (degree centrality, path length etc) with independently acquired maps of structural organization (cytoarchitecture, gene expression). The authors find several significant correlations between the primary functional gradient and the mentioned structural organizational principles. Finally, they connect these findings to the dual origin theory, and cognitive and behavioral scores.

One part that I did not find entirely convincing (or interesting) is the part about behavioral/cognitive prediction. The association found with grip strength seems tenuous. However

I am not even sure whether it is necessary or especially relevant to the journal. I think the authors actually agree with me cognition/behaviour is absent from the discussion.

Overall, this work is very interesting, thorough in its analysis and presents an interesting alternative perspective to the often suggested unimodal-to-multimodal FC axis of inter subject variability and nicely connects it to structural organizational principles.

Reproducibility and statistics seem appropriate.

The comments below are mostly about clarity of the presentation , which was sometimes lacking (with the exception of comment on line 236).

33: atypical or A typical? This is a really cumbersome way of defining FC. "Typically, FC is defined as the correlation of two element's BOLD time series."

39: not sure what is meant by "smoothing transition nature". However, I agree with first part.

50: "from" multiple perspectives

55: does this mean an "indicator"?

77-80: This needs to be split and reformulated. I (think I) understand the meaning from context but it is really hard to parse. Are there words missing?

86-88: Not sure what „with a with-in dataset repeated session“ is. Can you reformulate? If its too complicated maybe you can remove it here and more clearly explain in 133? Are you taking about the repeated sessions by by the HCP subjects (i.e. session on day 1 vs day 2)?

92: what are internal to outside cognitive processes? Maybe you can just remove it here

127: The inflated surface at the bottom right is not clearly labeled. As I understand it blue-to-red is defined by the first gradient from low ranking to high ranking. However, just above it says allocortex <—> neocortex. That is not based on the work you did before (diffusion map embedding etc), right? Also: isn't the orbito-frontal area also neocortex but low-ish ranking? Unless I misunderstand, this should probably be removed here until the association is shown later in the manuscript.

203: *have resulted

228-229: I think if you want to present the gradient-cognition association, it would be worth to explain more clearly, what you actually did methodologically. The present explanation will not allow for reproduction. I did not see any more specifics in the methods section.

235: Do you have an explanation for pain being so low on the gradient. I would expect it to be more to the „sensory“ side? Seems like a pretty big outlier.

236: I would like to see the effect of sex regressed out in the prediction of grip strength. Especially with grip strength, the confounding effect of sex can lead to misinterpretation. Just looking at the figure i think it is plausible that two big "blobs" (left female, right male) are present, and thus it is not just grip strength, but sex, that is predicted. Splitting grip strength prediction by sex might lead to two insignificant prediction (predicting once for males, once for females).

307-310: this is already said in line 245?

Reviewer #2 (Remarks to the Author):

Thank you for the opportunity to review “Cortical gradient of a functional similarity network captured by the geometry of cytoarchitectonic organization” by Meng and colleagues. The manuscript describes an extension of previous work looking at how functional and cytoarchitectonic gradients are correlated. The main novelty of the work is that the authors include multiple derivatives of fMRI time series, include ALFF, ReHo, etc. and apply the standard diffusion map embedding algorithm to recover gradients. They then show that these are often correlated with other structural gradients, and broadly resemble the Dual Origin pattern.

Overall, the paper is rigorous and comprehensive. The work builds on recent reports and will be of interest to that community. I do, however, have some suggestions for improvement:

We would like to thank the reviewer for the constructive assessments and valuable suggestions, we made a point-by-point response to all questions reviewer raised. All the modifications in the revised manuscript have been marked **yellow**.

Q1. The narrative is clear, but the writing is nevertheless awkward and needs work. For example:

“The primary gradient demonstrated layer-specific characteristics in microscopic gene expression and mesoscopic layer thickness, and was captured by geometric representation of a myelo- and cyto-architecture based laminar differentiation theorem, involving a dual origin theory.”

A: We rephrased this part for eliminating the confusion.

“The primary gradient demonstrated converging relationships with layer-specific microscopic gene expression and mesoscopic cortical layer thickness, and was captured by the geometric representation of a myelo- and cyto-architecture based laminar differentiation theorem, involving a dual origin theory.”

“decode ubiquitous cognitions”

A: We removed this part for eliminating the confusion.

“Atypical FC defines connectivity on the basis of the similarity between two elements’ dynamics”

A: We rephrased this sentence as reviewer 3 also mentioned.

“Typically, FC is defined as the correlation of two elements’ blood oxygen level-dependent (BOLD) time series.”

“continuous transition relationships”

A: We rephrased this part.

Increasing evidence has implied that a set of anatomically distributed functional systems/networks were anchored on the cerebral cortex by some axes describing the

spatially graded changes in the expression of connectivity patterns, which were the so-called “Gradients.”

Q2. In addition, the writing is often more complex than it needs to be. For example: “We then leveraged a canonical embedding algorithm based on manifold learning”. Surely the authors could have just said “dimensionality reduction” rather than “manifold learning”.

A: We rephrased this sentence as reviewer suggested.

“We then leveraged a canonical dimensionality reduction algorithm to map the primary gradient (Fig. 1).”

Q3. Importantly, the gradient derived by the authors (shown in Fig. 1) is also reminiscent of the well-known tSNR map for fMRI. Given that the gradient is derived entirely from fMRI measures, the authors should verify whether the two are correlated.

A: As the reviewer’s suggestion, we found a correlation between the tSNR map and the primary gradient. Moreover, we found that the tSNR map spatially correlated with all functional metrics maps except for the ReHo map (Table. S1), consistent with previous studies (Braga at al., 2017; Qing et al., 2019).

We also compared the primary gradient and tSNR map (both z-scored) across the sample of HCP-REST1 using paired *t*-test. The difference is widely spread across the cortex, including the occipital cortex, lateral temporal cortex, prefrontal cortex, orbitofrontal cortex, and insula (Fig. S5B, *right panel*). Although the primary gradient and the tSNR maps show a significant correlation ($r_s = -0.90$, Fig. S5B, *left panel*), the discrepancy suggests the primary gradient of the cortical functional similarity network is not simply dominated by tSNR.

We mentioned this in the revised manuscript, page 7, line 138 - 146:

*“We found a correlation ($r_s = 0.90$) between the temporal signal-to-noise ratio (tSNR) map and the primary gradient, Suppl. Fig. 5). Moreover, we found that the tSNR map spatially correlated with all functional metrics maps except for the ReHo map (Table S1), consistent with previous studies^{36,37}. We also compared the primary gradient and tSNR map (both z-scored) using paired *t*-test. The difference is widely spread across the cortex, including the occipital cortex, lateral temporal cortex, prefrontal cortex, orbitofrontal cortex, and insula (Suppl. Fig. 5B). Although the primary gradient and the tSNR maps show a significant correlation, the discrepancy suggests the primary gradient of the cortical functional similarity network is not simply dominated by tSNR.”*

Table. S1. Correlation between tSNR map with functional metrics

Correlation with tSNR	ALFF	fALFF	ReHo	DC	IEfficiency	gEfficiency	Path length
------	-------	------	----	-------------	-------------	-------------

Spearman r	-0.80	0.84	0.06	-0.44	-0.40	-0.41	0.42
p-value_{SAC}	< 0.0001	< 0.0001	0.65	0.018	0.023	0.027	0.03

Fig. S4. Relation with tSNR. (A) Group-mean tSNR map of HCR REST-1. **(B)** The correlation and differences (paired t -test, Bonferroni corrected) between primary gradient and tSNR across HCP-REST1 samples.

Refs:

- Braga, R. M. & Buckner, R. L. Parallel Interdigitated Distributed Networks within the Individual Estimated by Intrinsic Functional Connectivity. *Neuron* 95, 457-471.e5 (2017).
- Qing, Z. et al. The Impact of Spatial Normalization Strategies on the Temporal Features of the Resting-State Functional MRI: Spatial Normalization Before rs-fMRI Features Calculation May Reduce the Reliability. *Front. Neurosci.* 13, (2019).

Q4. “All the code is openly available at <https://github.com/YaoMeng94/FSN415> Gradient.” Contrary to this statement, there is no such repository and the code is not openly available.

A: We are sorry about the delay in uploading the codes and the wrong link to the repository. We have uploaded the codes and the correct link is <https://github.com/YaoMeng94/FSN-Gradient>

Q5. How were FC weights transformed to lengths for computing path lengths in the global efficiency measure?

A: We used the GRETNA toolbox (Wang, J. et al., 2015) to compute the network metrics including path lengths, the path lengths in the global efficiency are defined as the “harmonic mean” distance (Newman, 2003) between all possible node pairs and was computed by calling the functions from the MatlabBGL toolbox (version 4.0). Specifically, the path length between nodes i and j was defined as the sum of the edge lengths along the path, where each edge’s length was obtained by computing the reciprocal of the edge weight, $1/w_{ij}$. The shortest path length L_{ij} between nodes i and j and was defined as the length of the path with the shortest length between the two nodes. The characteristic path length L :

$$L = \frac{1}{1/(N(N-1)) \sum_{i=1}^N \sum_{j \neq i}^N 1/L_{ij}}$$

We also mentioned in the manuscript, page15, line 352-354:

“We used the GRETNA toolbox to compute multi-graph theoretical metrics, the detail computation process and formulas refer to Wang et al.”

Ref:

Wang, J. et al. GRETNA: a graph theoretical network analysis toolbox for imaging connectomics. *Front. Hum. Neurosci.* 9, (2015).

Newman, M. E. J. The Structure and Function of Complex Networks. *SIAM Rev.* 45, 167–256 (2003).

Q6. The fMRI measures chosen are somewhat arbitrary. What is the rationale for not including other commonly used measures, such as measures of variability (SD, MSSD, MSE, etc.)?

A: Our conception is that the brain’s functional topology can be largely represented by local and global metrics. And the ALFF is a close representation of variability features in the spectral domain (Zuo, 2010). We have constructed another primary gradient with extra variability metrics added, including SD and MSSD. The primary gradient remains the same spatial pattern as the main result and shows strong correlation with the one in the main result ($r_s = -0.99$, $p_{SAC} < 0.0001$, Fig. S6).

Fig. S5. The effects of variability metrics. The primary gradient constructed with extra variability metrics (including SD and MSSD) added and the correlation with the original primary gradient.

Ref:

Zuo, X.-N. A Note on Measures of Single Timeseries Activity in Resting-State fMRI Studies. *Nat Prec* (2010) doi:10.1038/npre.2010.4379.1.

Reviewer #3 (Remarks to the Author):

This work connects macroscopic functional connectivity organizational principles derived from a variety of FC measures (ReHo, ALFF, fALLF) as well as graph theoretical measures (degree centrality, path length etc) with independently acquired maps of structural organization (cytoarchitecture, gene expression). The authors find several significant correlations between the primary functional gradient and the mentioned structural organizational principles. Finally, they connect these findings to the dual origin theory, and cognitive and behavioral scores.

One part that I did not find entirely convincing (or interesting) is the part about behavioral/cognitive prediction. The association found with grip strength seems tenuous. However I am not even sure whether it is necessary or especially relevant to the journal. I think the authors actually agree with me cognition/behaviour is absent from the discussion.

Overall, this work is very interesting, thorough in its analysis and presents an interesting alternative perspective to the often suggested unimodal-to-multimodal FC axis of inter subject variability and nicely connects it to structural organizational principles.

Reproducibility and statistics seem appropriate.

We would like to thank the reviewer for the constructive assessments and valuable suggestions, we made a point-by-point response to all questions the reviewer raised. All the modifications in the revised manuscript have been marked **yellow**.

And as the reviewer mentioned that the **behavioral/cognitive** part is disconnected from the other sections, we have decided to remove this part from the manuscript.

The comments below are mostly about clarity of the presentation, which was sometimes lacking (with the exception of comment on line 236).

33: atypical or A typical? This is a really cumbersome way of defining FC. "Typically, FC is defined as the correlation of two element's BOLD time series."

A: It should be "A typical". And we have rephrased this part according to the reviewer's suggestion.

"Typically, FC is defined as the correlation of two elements' blood oxygen level-dependent (BOLD) time series."

39: not sure what is meant by "smoothing transition nature". However, I agree with first part.

A: We rephrased this sentence for clearer narration.

"However, the FC itself cannot provide organizing principles of cortical topology."

50: "from" multiple perspectives

A: We have corrected this typo in the revised manuscript.

... to describe the spatial organization of the cerebral cortex from multiple perspectives.

55: does this mean an "indicator"?

A: Yes. And we have corrected this typo in the revised manuscript.

"... local activities and global communication indicators."

77-80: This needs to be split and reformulated. I (think I) understand the meaning from context but it is really hard to parse. Are there words missing?

A: We have rephrased this part.

"There have been some spatially more detailed measures, such as myeloarchitecture, cytoarchitecture, and cortical-cortical connections from tract tracings. This converging evidence has been summarized as an evolutionary, developmental, converged cortical structural organization principle, the dual origin theory, which has not been connected to macroscale functional topology."

86-88: Not sure what, with a with-in dataset repeated session" is. Can you reformulate? If its too complicated maybe you can remove it here and more clearly explain in 133? Are you talking about the repeated sessions by the HCP subjects (i.e. session on day 1 vs day 2)?

A: Yes, the "with-in dataset repeated session" means the sessions day1 vs day2. The first session of HCP data, HCP-REST1, was used to conduct the main analysis. The second session, HCP-REST2, was used to test the reproducibility of the primary gradient as internal validation. We have changed the overall expression of the validation part and reformulated multiple sentences for clearer narration, including

Page 5 line 99, **Results 2.1:**

"All the results presented in the main text is based on the data of first session of HCP (HCP-REST1)."

Page 7 line 134, **Results 2.1:**

"It could be reproduced in internal validation (HCP-REST2) ($r_s = 1$, $p_{SAC} < 0.0001$) and external validation (independent MSC dataset) ($r_s = 0.53$, $p_{SAC} = 0.0006$) analysis (Suppl. Fig. 1)."

Page 15 line 315, **Materials and Methods 4.1:**

"We used data of both two runs in HCP-REST1 session to conduct the main analysis."

"In addition, we used the data of both two runs in HCP-REST2 session as internal validation to test the reproducibility of the primary gradient."

Page 15 line 319, **Materials and Methods 4.1:**

"We used an independent dataset-Midnight Scan Club (MSC) as the external validation of the proposed functional similarity network encoded multifaced gradient."

92: what are internal to outside cognitive processes? Maybe you can just remove it here

A: We have removed this sentence since we decided not to include the cognition part in this manuscript as the reviewer suggested.

Finally, the primary gradient also associated with a cognitive function axis, which represented internal to outside cognitive processes.

127: The inflated surface at the bottom right is not clearly labeled. As I understand it blue-to-red is defined by the first gradient from low ranking to high ranking. However, just above it says allocortex \longleftrightarrow neocortex. That is not based on the work you did before (diffusion map embedding etc), right? Also: isn't the orbitofrontal area also neocortex but low-ish ranking? Unless I misunderstand, this should probably be removed here until the association is shown later in the manuscript.

A: Yes, the bottom right brain renders in Figure 1. is displaying the primary gradient. And the allocortex \longleftrightarrow neocortex label is based on the latter part of this manuscript. The orbitofrontal cortex is characterized neocortex, but its spatial location is close to two allocortex origins. In our geometry model of dual origin theory, the orbitofrontal cortex locates in the low part of the axis. That may explain its lowish ranking in the primary gradient. We have removed the allocortex \longleftrightarrow neocortex label in the figures (Fig. 1 & Fig. 2) before the corresponding section as the reviewer suggested.

Figure 1. Schematic diagram of the multifaceted functional gradient. The top panel shows seven metrics derived from the blood oxygen level-dependent signal. For each individual, all seven metrics maps vectorized to form a feature matrix, and the normal angle of each pair of brain areas or region of interest was calculated as a similarity measure (middle panel). A symmetric similarity matrix resulted from preceding procedures and fed into a diffusion map embedding algorithm, to project the high dimensional similarity profile to a series of low dimensional embeddings. The first component of embeddings was selected as the primary gradient because it better explained the half variance of the input similarity matrix, and was then rendered in the bottom row on the inflated surface (bottom panel).

Figure 2. Space distribution of the multifaceted functional gradient. (A) The first component of embeddings, namely the primary gradient, was rendered on the inflated surface (left) and unfolded flat surface (right). **(B)** According to the Mesulam laminar differentiation class atlas, the primary gradient was classified as a different class based on the corresponding spatial location.

203: *have resulted

A: We have corrected this typo in the revised manuscript.

... have resulted in a comprehensive framework ...

228-229: I think if you want to present the gradient-cognition association, it would be worth to explain more clearly, what you actually did methodologically. The present explanation will not allow for reproduction. I did not see any more specifics in the methods section.

A: As the reviewer mentioned previously, we have decided to remove the gradient-cognition part to focus more on the association between the microstructural principles and the primary functional gradient. In the previous manuscript, we explored the gradient-cognition

235: Do you have an explanation for pain being so low on the gradient. I would expect it to be more to the “sensory” side? Seems like a pretty big outlier.

A: It seems that the pain map from the NeuroSynth database mainly locates the anterior cingulate cortex and anterior insula in which these areas also showed at the low part of the primary gradient. This situation may lead to the pain located at the low part of the gradient-cognition axis.

236: I would like to see the effect of sex regressed out in the prediction of grip strength. Especially with grip strength, the confounding effect of sex can lead to misinterpretation. Just looking at the figure i think it is plausible that two big "blobs" (left female, right male) are present, and thus it is not just grip strength, but sex, that is predicted. Splitting grip strength prediction by sex might lead to two insignificant prediction (predicting once for males, once for females).

A: As mentioned previously, we have removed the cognition part. We agreed with the reviewer this work's focus is on the association between functional gradient and meso- and micro-scale structural features. So, we make the decision to remove the cognition part from the manuscript.

307-310: this is already said in line 245?

A: We have removed this sentence in the revised manuscript.

~~The proposed gradient showed cytoarchitectonic cortex layer specific characteristics in microscopic gene expressions and mesoscopic layer thicknesses, and more importantly, was ideally situated in a myelo and cytoarchitecture informed laminar differentiation stream theorem, which encompassed the dual origin theory.~~

REVIEWERS' COMMENTS:

Reviewer #2 (Remarks to the Author):

The authors have comprehensively addressed my concerns and I recommend publication.

Reviewer #3 (Remarks to the Author):

I think the authors for addressing my concerns and I think that removing the section on behavior made sense. Overall, my concerns have been addressed.
I wish the authors all the best.